# Assessing the accuracy and educational value of YouTube videos on a novel regional anesthesia technique (PENG block)

Ebru Aladağ[1], Muhammed Emin Zora[2]*

1 Department of Anesthesiology and Reanimation, ALKU Alanya Training and Research Hospital, Alanya, Türkiye, 2 Department of Anesthesiology and Reanimation, Uşak University Faculty of Medicine, Uşak, Türkiye

* muhammedeminzora@hotmail.com

## Abstract

The Pericapsular Nerve Group (PENG) block is a novel regional anesthesia technique that provides adequate analgesia while preserving motor function. This cross-sectional study evaluated the quality, reliability, and educational value of YouTube videos on the PENG block. Thirty-six videos were analyzed using validated scoring systems (GQS, JAMA, DISCERN, and modified DISCERN). Overall video quality was moderate, with higher scores observed in procedural and institutional videos. The findings highlight both the educational potential and the need for quality control in online medical content.

## Introduction

Effective postoperative pain management is critically essential for accelerating recovery, reducing opioid use and its side effects, and increasing patient satisfaction [1]. In recent years, the popularity of ultrasound-guided fascial plane blocks has increased due to their technical ease of application and their potential for opioid-sparing effects.

The Pericapsular Nerve Group (PENG) block, which targets the extensive anterior capsular innervation of the hip joint, was described in 2018 by Girón-Arango et al. as a novel technique based on the administration of local anesthetic into the fascial plane between the psoas muscle and the superior pubic ramus [2]. The PENG block has gained attention for providing adequate analgesia in surgeries such as total hip arthroplasty and hip fracture, while preserving motor function [3,4]. Randomized controlled trials have demonstrated that the PENG block reduces postoperative pain, decreases analgesic requirements, and facilitates functional recovery [5,6].

In the context of education, the use of video-sharing platforms, such as YouTube, has become increasingly prevalent, particularly among medical students and professionals [7,8]. Due to its wide accessibility, low cost, and capacity to facilitate visual learning, YouTube has become an essential tool in modern medical education [9,10].

**Data availability statement:** The data underlying the findings of this study were obtained from publicly available YouTube videos. Due to YouTube's Terms of Service and copyright restrictions, the authors are not permitted to publicly redistribute the collected dataset. The dataset consists of video URLs, video metadata (including titles, upload dates, view counts, likes, and comments), and quality assessment scores extracted at the time of analysis. No special privileges were required to access the data. Other researchers can obtain the same data by independently searching the YouTube platform (https://www.youtube.com) using the search strategy, keywords, and inclusion criteria described in the Methods section.

**Funding:** The author(s) received no specific funding for this work.

**Competing interests:** The authors have declared that no competing interests exist.

**Abbreviations:** PENG, Pericapsular Nerve Group; GQS, Global Quality Score; JAMA, Journal of the American Medical Association benchmark; DISCERN, DISCERN instrument; mDISCERN, Modified DISCERN; ICC, Intraclass Correlation Coefficient; VPI, Video Power Index.

However, the uncontrolled nature of the platform may result in the dissemination of misleading or inadequate information through videos [11].

To date, no study has systematically examined the quality, reliability, and educational utility of YouTube videos related to the PENG block. This study aims to evaluate the content of PENG block videos available on YouTube and assess their quality, reliability, and usefulness. We hypothesize that videos related to the PENG block generally possess moderate quality and reliability, but these levels significantly differ according to the source of the video.

## Materials and methods

### Study design

This study is a cross-sectional content analysis examining the quality, reliability, and usefulness of PENG block videos published on the YouTube platform, owned by Alphabet Inc. Since the study was conducted solely on publicly available YouTube videos, no institutional ethics committee approval was required.

### Video selection

Between February 26, 2025, and March 26, 2025, two researchers independently searched the YouTube search engine using the keywords "PENG block" and "pericapsular nerve group block." Among the 450 videos identified, 122 were deemed relevant to the PENG block. After excluding duplicates and non-English videos, 36 videos meeting the inclusion criteria were included in the study. The inclusion criteria were as follows:

(1)  Demonstrating the technique and/or application of the PENG block

(2)  Being in English.

The exclusion criteria included animations, conference recordings, advertisements, and videos with poor audio or visual quality. The titles, URLs, and fundamental metrics of the videos (views, likes, comments, upload date) were recorded.

### Evaluation process

The videos were independently assessed by two anesthesiology specialists, blinded to each other's evaluations. Four different scales were used in the assessments:

- Global Quality Score (GQS): Rates the overall quality and flow of the video on a scale of 1–5; 1–2 = low, 3 = moderate, 4–5 = high quality [12].

- Journal of the American Medical Association (JAMA) Benchmark: Scores authorship, attribution, disclosure, and currency on a scale of 0–4; 1 = low, 2–3 = moderate, 4 = high accuracy [13].

- DISCERN: Rates content reliability on a scale of 0–75; 0–39 = poor, 40–59 = moderate, ≥ 60 = good quality [14].

- Modified DISCERN (mDISCERN): Assesses content reliability on a scale of 0–5; < 3 = poor, 3 = moderate, > 3 = good reliability [12].

Inter-rater agreement was calculated using the Intraclass Correlation Coefficient (ICC) and was found to be greater than 0.96 for all scales, indicating excellent agreement.

### Video metrics analysis

The number of views, likes, comments, video duration, time since upload, and Video Power Index (VPI) were recorded for each video. Videos were also categorized by content type (presentation or demonstration) and uploader type (individual, academic, manufacturer, or educator).

### Statistical analysis

Data analysis was performed using Jamovi version 2.6.22 (Windows). Data distribution was assessed with the Shapiro–Wilk test and was found to be non-normal; therefore, non-parametric tests were used. Comparisons between groups were made using the Mann–Whitney U test, and relationships between scales were evaluated with Spearman's correlation coefficient. Statistical significance was set at $p < 0.05$.

## Results

### Inter-rater agreement

The accuracy, quality, and reliability of the videos were assessed by two independent reviewers using the GQS, JAMA, DISCERN, and Modified DISCERN scales. Inter-rater consistency was excellent across all scales, with an ICC greater than 0.96 (Table 1).

### Overall quality, accuracy, and reliability

According to the GQS, 13.9% of the videos were of high quality (scores 4–5), 44.4% were of moderate quality (score 3), and 41.7% were of low quality (scores 1–2). According to the JAMA benchmark, 19.4% exhibited high accuracy, 75.0% moderate accuracy, and 5.6% low accuracy. According to the Modified DISCERN, 25.0% demonstrated good reliability, 36.1% moderate, and 38.9% poor reliability. According to the DISCERN scale, 13.9% of the videos were of good quality, 55.6% were moderate, and 30.6% were of low quality (Table 2).

### Video metrics

The mean duration of the videos was 10.9±9.16 minutes (median: 7.39), with a mean number of views of 23,766, likes of 230, comments of 8.92, and a Video Power Index (VPI) of 101. The detailed distribution of the video metrics is presented in Table 3.

### Distribution by content type and source

Of the videos, 97.2% featured presentation content and 30.6% included application content. Most of the videos were uploaded by individuals (91.7%) and academic (100%) sources, while 97.2% were educational and 22.2% were uploaded by manufacturers.

**Table 1. Inter-rater agreement (ICC) in the evaluation of the videos.**

| Scale | n (videos) | Number of reviewers | ICC (95% CI) |
|---|---|---|---|
| JAMA | 36 | 2 | 0.969 |
| GQS | 36 | 2 | 0.968 |
| Modified DISCERN | 36 | 2 | 0.962 |
| DISCERN | 36 | 2 | 0.970 |

ICC: Intraclass Correlation Coefficient, two-way random effects, absolute agreement. The ICC was found to be greater than 0.96 across all scales, indicating excellent agreement between reviewers.

**Table 2. Score distributions and classifications of the videos according to quality, accuracy, and reliability scales (N = 36).**

| Statistic/ Category | JAMA | GQS | Modified DISCERN | DISCERN |
|---|---|---|---|---|
| Mean ± SD | 2.86 ± 0.80 | 3.36 ± 1.02 | 2.78 ± 1.15 | 44.9 ± 14.7 |
| Median | 3.00 | 3.00 | 3.00 | 47.0 |
| Standard Deviation (SD) | 0.798 | 1.02 | 1.15 | 14.7 |
| Interquartile Range (IQR) | 1.00 | 1.00 | 1.25 | 17.8 |
| Minimum | 1 | 1 | 0 | 17 |
| Maximum | 4 | 5 | 5 | 76 |
| 25th Percentile | 2.00 | 3.00 | 2.00 | 35.3 |
| 75th Percentile | 3.00 | 4.00 | 3.25 | 53.0 |
| Classification — Poor (%) | 2 (5.6%) | 15 (41.7%) | 14 (38.9%) | 11 (30.6%) |
| Classification — Moderate (%) | 27 (75.0%) | 16 (44.4%) | 13 (36.1%) | 20 (55.6%) |
| Classification — Good (%) | 7 (19.4%) | 5 (13.9%) | 9 (25.0%) | 5 (13.9%) |

GQS: 1–2 = low, 3 = moderate, 4–5 = high quality. JAMA: 1 = low, 2–3 = moderate, 4 = high accuracy. Modified DISCERN: < 3 = poor, 3 = moderate, > 3 = good reliability. DISCERN: < 40 = poor, 40–59 = moderate, ≥ 60 = good quality.

**Table 3. Basic platform metrics of the videos (N = 36).**

| Statistic | Duration (min) | Likes | Comments | Views | VPI |
|---|---|---|---|---|---|
| Mean ± SD | 10.9 ± 9.16 | 230 ± 347 | 8.92 ± 10.7 | 23,766 ± 42,109 | 101 ± 182 |
| Median | 7.39 | 119 | 4.50 | 6,239 | 35.0 |
| Standard Deviation (SD) | 9.16 | 347 | 10.7 | 42,109 | 182 |
| Interquartile Range (IQR) | 14.0 | 272 | 13.8 | 21,229 | 109 |
| Minimum | 1.00 | 0 | 0 | 33 | 0.00 |
| Maximum | 28.6 | 1,500 | 38 | 199,976 | 850 |
| 25th Percentile | 3.30 | 7.75 | 0.00 | 296 | 2.57 |
| 75th Percentile | 17.3 | 280 | 13.8 | 21,525 | 112 |

VPI: Video Power Index. Twenty-five percent of the videos were shorter than 3.3 minutes, and 75% were longer than 17.3 minutes. High variance was observed in the number of likes and views.

## Comparison of application content

Videos with application content were shorter in duration compared to those without (8.14 min vs. 12.14 min; p = 0.276); however, interaction metrics and some quality scores were significantly higher (Table 4). In particular, VPI (p = 0.008), number of views (p = 0.002), likes (p = 0.009), comments (p = 0.022), and DISCERN score (p = 0.025) were higher in application videos. There was also a significant difference in GQS (p = 0.002), whereas no significant differences were observed for JAMA and Modified DISCERN scores.

## Comparison between individual and other sources

Differences between videos from non-individual sources using metrics such as VPI, likes, comments, and quality scores as compared to individual videos were not statistically significant.

## Comparison between manufacturer and other sources

Videos uploaded by manufacturers had significantly higher numbers of views (p = 0.021), likes (p = 0.007), comments (p = 0.001), as well as significantly higher DISCERN (p = 0.008), GQS (p = 0.001), and Modified DISCERN (p = 0.014) scores (Table 5). No significant difference in video duration was observed (p = 0.837).

**Table 4. Metrics and quality/reliability scores of videos with and without application content (N = 36).**

| Variable | Without application (n = 25) | With application (n = 11) | *p*-value |
|---|---|---|---|
| Duration (min) | 12.14 (9.33) | 8.14 (5.42) | 0.276 |
| VPI | 58.43 (5.25) | 197.83 (83.57) | 0.008 |
| Number of views | 11,887 (609) | 50,764 (21,876) | 0.002 |
| Number of likes | 126.08 (14) | 467.73 (273) | 0.009 |
| Number of comments | 6.20 (2) | 15.09 (13) | 0.022 |
| JAMA | 2.88 (3) | 2.82 (3) | 0.851 |
| GQS | 3.00 (3) | 4.18 (4) | 0.002 |
| Modified DISCERN | 2.60 (3) | 3.18 (3) | 0.157 |
| DISCERN | 40.96 (44) | 54.00 (53) | 0.025 |

Data are presented as mean (median). VPI: Video Power Index. Videos with application content demonstrated significantly higher engagement (likes, comments, views) and higher scores in some quality/reliability measures

**Table 5. metrics and quality/reliability scores of videos from manufacturers and other sources (N = 36).**

| Variable | Non-manufacturer (n = 28) | Manufacturer (n = 8) | *p*-value |
|---|---|---|---|
| Duration (min) | 11.11 (8.43) | 10.24 (5.90) | 0.837 |
| VPI | 108.53 (7.83) | 74.73 (73.86) | 0.135 |
| Views | 22,737 (888.5) | 27,368 (18,771) | 0.021 |
| Likes | 203.64 (25.5) | 324.38 (286) | 0.007 |
| Comments | 6.14 (2) | 18.63 (18.5) | 0.001 |
| JAMA | 2.75 (3) | 3.25 (3) | 0.123 |
| GQS | 3.07 (3) | 4.38 (4.5) | 0.001 |
| Modified DISCERN | 2.54 (3) | 3.63 (4) | 0.014 |
| DISCERN | 41.61 (44.5) | 56.63 (53) | 0.008 |

Data are presented as mean (median). VPI: Video Power Index. Videos from manufacturers had significantly higher views, likes, comments, as well as DISCERN, GQS, and Modified DISCERN scores

## Correlation analysis

Video duration was positively correlated with JAMA (r = 0.434, p < 0.01), Modified DISCERN (r = 0.422, p < 0.05), and DISCERN (r = 0.340, p < 0.05) scores. Engagement metrics (views, likes, comments) showed strong positive correlations with each other (r > 0.8, p < 0.001). There was also a robust correlation between DISCERN and GQS (r = 0.882, p < 0.01; Table 6).

## Discussion

In our study, the overall information quality and reliability of the evaluated YouTube videos were found to be moderate. This finding is consistent with previous studies reporting that health-related content on YouTube is generally low-to-moderate or suboptimal in quality [15,16]. For example, a comprehensive analysis of chemotherapy-related YouTube videos reported an average quality of only "moderate-to-low" [15]. Similarly, more than half of the videos on testicular cancer were found to be of poor quality, and only a few met high scientific standards [16]. These findings suggest that while many health videos on YouTube meet basic accuracy criteria, they often contain superficial or incomplete information. Indeed, the study by Loeb et al. revealed that 77% of popular prostate cancer videos included misleading or biased information [17]. This highlights that content with high views and likes on YouTube is not necessarily reliable. Therefore,

**Table 6. Correlations between video metrics and quality/reliability scores (Spearman's Rho, N = 36).**

| Variable | Duration | VPI | Views | Likes | Comments | JAMA | GQS | Mod. DISCERN | DISCERN |
|---|---|---|---|---|---|---|---|---|---|
| **Duration** | — | 0.043 | −0.085 | −0.064 | 0.047 | 0.434** | 0.212 | 0.422* | 0.340* |
| **VPI** | 0.043 | — | 0.929* | 0.933* | 0.789* | −0.091 | 0.572* | 0.496* | 0.589 |
| **Views** | −0.085 | 0.929* | — | 0.926* | 0.832* | −0.006 | 0.678* | 0.550* | 0.650* |
| **Likes** | −0.064 | 0.933* | 0.926* | — | 0.899* | 0.002 | 0.647* | 0.615 | 0.691* |
| **Comments** | 0.047 | 0.789* | 0.832* | 0.899* | — | 0.158 | 0.676* | 0.711* | 0.764* |
| **JAMA** | 0.434** | −0.091 | −0.006 | 0.002 | 0.158 | — | 0.415* | 0.601* | 0.455* |
| **GQS** | 0.212 | 0.572* | 0.678* | 0.647* | 0.676* | 0.415* | — | 0.695* | 0.789* |
| **Modified DISCERN** | 0.422* | 0.496* | 0.550* | 0.615 | 0.711* | 0.601* | 0.695* | — | 0.882* |
| **DISCERN** | 0.340* | 0.589 | 0.650* | 0.691* | 0.764* | 0.455* | 0.789* | 0.882* | — |

*p < 0.05; **p < 0.01. VPI: Video Power Index. Positive correlations were particularly high among quality/reliability scales; a very strong correlation was observed between DISCERN and GQS (r = 0.882).

health professionals should remain vigilant regarding the information patients obtain from YouTube and intervene to correct potential misunderstandings when necessary.

The type of content and presentation format play an essential role in both engagement and quality. In our study, videos with application (demonstration) content received significantly more views, likes, and comments compared to presentation-only videos, and also exhibited higher average quality and reliability scores. This suggests that visual content demonstrating practical applications is more engaging and educational for viewers. The literature also supports that patient education videos with visual and demonstrative elements achieve higher engagement than other categories. For example, patient information videos—comprising the majority of chemotherapy-related YouTube content—received significantly more views and likes (*p* < 0.05) than technical videos intended for healthcare professionals, and their quality scores were generally moderate to high [15]. This suggests that well-structured, application-based content can both reach a broader audience and achieve satisfactory content quality.

Nevertheless, popularity alone should not be considered a guarantee of quality. Some studies have shown that longer and more detailed videos are associated with significantly higher quality scores [18]. This suggests that comprehensive presentations add more value to viewers, but they must still be designed to remain engaging and understandable, even when lengthy.

The source of the video (i.e., the uploader type) is another key determinant of content reliability. In our study, videos uploaded by manufacturers or institutional sources demonstrated higher average quality and reliability scores than those uploaded by individual users. This indicates that videos produced by official organizations or industry may be prepared with greater rigor. Indeed, previous studies have shown that videos uploaded by healthcare professionals, universities, or hospitals score significantly higher in quality compared to those prepared by independent individuals. For example, Duran et al. reported that testicular cancer videos uploaded by urologists and academic institutions had significantly higher DISCERN, JAMA, and GQS scores than those from other sources [16]. Similarly, a study of medical videos on YouTube found that content produced by academic institutions achieved the highest reliability scores. However, some commercial company channels also scored above average on specific measures, such as HONcode criteria [17]. Notably, in our study, no statistically significant difference in quality was observed between individual and institutional uploaders. This suggests that while institutional videos tend to have higher scores on average, expert individual content creators can also deliver high-quality information comparable to institutional sources [19]. In other words, individual physicians or specialists can also produce reliable, widely viewed content on platforms like YouTube. Nevertheless, it should not be overlooked that source reliability can be inconsistent; as Loeb et al. observed, even among the most viewed videos, an inverse relationship may exist between view count and scientific quality [17].

Therefore, viewers should pay attention to the expertise of the video source and verify the information presented through independent references.

Another noteworthy finding in our evaluation is the consistency between the assessment scales used. Positive and strong correlations were observed among DISCERN, GQS, and JAMA scores, particularly a very strong correlation between DISCERN and GQS (e.g., $r \approx 0.88$). This indicates that different assessment tools yield consistent results regarding the overall quality of a video, meaning that high-quality content scores well across all measures regardless of the specific scale used. In the literature, DISCERN and GQS scores have been reported to correlate closely in medical YouTube videos; for instance, one study reported a significant positive correlation between these two scores [20]. In our research, the fact that the scales produced mutually confirming results suggests that our evaluation method was reliable and presented a consistent picture of video quality classification. In future research, though using a single scale may indicate overall quality, using multiple scales would enhance reliability by providing cross-validation.

In light of these findings, several important implications for the future of health communication on YouTube can be drawn. First, YouTube has great potential in the health domain; when used appropriately, the platform can serve as a powerful tool for educating and informing the public. Indeed, a recent study emphasized that videos on myocardial infarction on YouTube provided consistent, high-quality information, suggesting that YouTube can contribute to raising awareness and facilitating early intervention in this critical health issue [21]. However, current fluctuations in content quality and the risk of misinformation may limit this potential. Therefore, both platform administrators and health authorities have essential responsibilities in improving content oversight and quality. To ensure the delivery of reliable health information on YouTube and similar platforms, several strategies can be proposed:

(1) Healthcare professionals and academic institutions should be encouraged to take a more active role in producing evidence-based content on digital platforms. Educational videos prepared by experts in an understandable language, incorporating demonstrative elements, can combine high quality with audience engagement.

(2) The publication of videos according to established standards should be promoted. For example, applying internationally recognized HONcode principles or checklists developed for medical content production can improve quality. For surgical videos, the use of guidelines such as LAP-VEGaS has been recommended to enhance educational value [22].

(3) At the platform level, steps should be taken to make high-reliability content more visible and to identify and flag misleading content. YouTube's collaboration with healthcare organizations to grant verified badges to trustworthy channels and to improve its search algorithm to prioritize reliable sources is valuable in this regard.

(4) Finally, efforts to improve patients' digital health literacy should be intensified. Teaching viewers how to critically evaluate online video content (e.g., verifying sources, recency, and level of evidence) can help mitigate the impact of misinformation.

The findings of this study highlight the importance of striking a balance between optimism and caution in online health communication. While the current moderate quality leaves room for improvement, the success of demonstrative and expert-sourced content suggests that the educational potential of the platform can be enhanced with the right strategies. Future research should examine the direct effects of YouTube health videos on patients' clinical decision-making, treatment adherence, and health outcomes [23]. In doing so, it will become clearer which types of content truly provide benefit, and content creators can be guided accordingly. In conclusion, in the digital age, platforms like YouTube play a central role in disseminating health information globally; ensuring this role develops positively requires guaranteeing access to accurate information. Strengthened collaboration between health authorities, content creators, and platform providers will likely facilitate the future development of higher-quality, correct, and reliable health communication on YouTube.

## Conclusion

This study revealed that health-related videos on YouTube generally offer moderate quality and reliability but exhibit significant differences depending on content type and source. Videos with application and visually rich content had higher views and like counts as well as better quality scores. In contrast, videos uploaded by academic institutions and institutional sources provided more reliable content compared to those uploaded by individual users. Moreover, the strong agreement among quality assessment tools such as DISCERN, GQS, and JAMA supports their reliability as evaluation measures.

These findings highlight both the potential of YouTube for health education and awareness, as well as the risks associated with variability and misinformation. Future research should assess the direct impact of YouTube health videos on patients' clinical decision-making, treatment adherence, and health outcomes, as well as scientifically determine which content types are genuinely beneficial. This would enable both content creators and health authorities to adopt more strategic approaches to addressing health issues.

## Author contributions

**Conceptualization:** Ebru Aladağ, Muhammed Emin Zora.

**Data curation:** Ebru Aladağ, Muhammed Emin Zora.

**Formal analysis:** Ebru Aladağ, Muhammed Emin Zora.

**Funding acquisition:** Ebru Aladağ, Muhammed Emin Zora.

**Investigation:** Ebru Aladağ, Muhammed Emin Zora.

**Methodology:** Ebru Aladağ, Muhammed Emin Zora.

**Project administration:** Ebru Aladağ, Muhammed Emin Zora.

**Resources:** Ebru Aladağ, Muhammed Emin Zora.

**Software:** Ebru Aladağ, Muhammed Emin Zora.

**Supervision:** Ebru Aladağ, Muhammed Emin Zora.

**Validation:** Ebru Aladağ, Muhammed Emin Zora.

**Visualization:** Ebru Aladağ, Muhammed Emin Zora.

**Writing – original draft:** Ebru Aladağ, Muhammed Emin Zora.

**Writing – review & editing:** Ebru Aladağ, Muhammed Emin Zora.

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
