## [Decision Letter · Decision Letter 0]

12 Dec 2025

Dear Dr. Zora,

Thank you for submitting your manuscript to PLOS ONE. After careful consideration, we feel that it has merit but does not fully meet PLOS ONE’s publication criteria as it currently stands. Therefore, we invite you to submit a revised version of the manuscript that addresses the points raised during the review process.

We look forward to receiving your revised manuscript.

Kind regards,

Cheong Kim

Academic Editor

PLOS One

**Journal Requirements:**

3. In the online submission form you indicate that your data is not available for proprietary reasons and have provided a contact point for accessing this data. Please note that your current contact point is a co-author on this manuscript. According to our Data Policy, the contact point must not be an author on the manuscript and must be an institutional contact, ideally not an individual. Please revise your data statement to a non-author institutional point of contact, such as a data access or ethics committee, and send this to us via return email. Please also include contact information for the third party organization, and please include the full citation of where the data can be found.

Reviewers' comments:

Reviewer's Responses to Questions

**Comments to the Author**

1. Is the manuscript technically sound, and do the data support the conclusions?

Reviewer #1: Yes

Reviewer #2: Yes

2. Has the statistical analysis been performed appropriately and rigorously?

Reviewer #1: No

Reviewer #2: Yes

3. Have the authors made all data underlying the findings in their manuscript fully available?

Reviewer #1: Yes

Reviewer #2: Yes

4. Is the manuscript presented in an intelligible fashion and written in standard English?

Reviewer #1: Yes

Reviewer #2: Yes

Reviewer #1: This is such a captivating topic. I also love the fact that the research article was concise & easy to read and understand. The tables being in an appendix made reading through the article smooth and uninterrupted. The research was well done & the scoring systems correlated in the results. The were also no plagiarism noted.

However, there are a few things I would like to highlight;

1. Ensure to use the same wordings in the text as was used in the table to allow for uniformity. In the results section, regarding reference to the discern scale, correct "55.6% were mild" to "55.6% were moderate.

2. In the comparison between individuals and other sources; correct that paragraph to "Differences between videos from non individual sources using metrics such as VPI, likes, comments and quality scores as compared to individual videos were not statistically significant."

3. In the discussion section; 1st paragraph, moderate quality does not align with poor quality, find a correct correlation or you could rephrase the sentence so as to align with your results.

4. In the last line of the 4th paragraph, it should be preferably put as "In future research, though using a single scale may provide an indication of overall quality, using multiple scales would enhance reliability by providing cross validation.

Lastly, I noticed that the percentage figures quoted for JAMA, GQS and Modified Discern scales in the overall quality, accuracy and reliability portion of the results section didn't tally with what was on the table. Only those for the Discern scale were in tandem. Kindly look in and reassess your data.

Thank you and well done

Reviewer #2: . Abstract:

Result: Clarify the first sentence for precision. Instead of "Out of 422 patients, 37.9% of patients developed about 214 complications," consider:

"Of the 422 patients, 160 (37.9%) experienced one or more critical events, with a total of 214 complications recorded."

Conclusion: Ensure consistency. The phrase "unhealthy body weight" in the conclusion should be explicitly tied to the BMI categories used (overweight/underweight).

2. Introduction:

Flow: Improve connectivity between sentences. Use transition words (e.g., Furthermore, Consequently, However) to guide the reader through the rationale.

Citations: Verify that the statement about limited data in Africa is directly supported by the newly added references (e.g., Blaise Pascal FN et al. 2021).

3. Methods:

Ethics: The approval code R/C/S/D/357/01/18 is now included, which is good.

Clarity: The explanation for not analyzing Clavien-Dindo Grade III-V separately (sample size limitations) is acceptable but should be stated clearly in the limitations section if not already.

4. Results:

Language: Replace informal phrases. For example:

"Even though it’s not surprising..." → "As anticipated, the incidence..."

"The proportions... were nearly equal..." → "The proportions were similar across groups, ranging from 35% to 38.4%."

Narrative: When discussing Table 5 in the text, explicitly state the key findings (e.g., "Patients attended by untrained nurses had over three times the odds of experiencing a critical event (AOR=3.15...).")

5. Discussion:

Tone: Maintain an objective, scientific tone. Avoid speculative phrasing like "We couldn’t provide the possible reasons..." Instead, state: "The reason for this discrepancy is not clear from our data but may relate to differences in case mix or definitions."

Structure: When comparing studies, first state the agreement or disagreement, then provide the comparative data, and finally offer a brief, plausible reason for the difference.

6. Language & Grammar (Final Polish):

Conduct a meticulous line-by-line edit focusing on:

Article Use: Ensure correct use of "a," "an," and "the."

Prepositions: Check "in," "on," "at," "for," etc.

Subject-Verb Agreement: (e.g., "The data were analyzed").

Plurals: (e.g., "complications").

Typos: "Sever hypoxemia" → "Severe hypoxemia"; "statistically significant" (not "statically").

7. Figures & Tables:

Flowchart (Figure 1): Ensure it matches the description in the "Response to Reviewers":

Quantify excluded patients (n=11).

Place "n = 422" label next to "Patients included in the study".

Final branch clearly shows: "Patients with ≥1 critical event (n=160)" and "Patients without critical events (n=262)".

Submission Checklist:

Before final submission, verify:

The Response to Reviewers letter is complete, polite, and addresses every point raised by each reviewer and the editor.

The Data Availability Statement in the submission system is accurate.

All supporting files (dataset, checklist, ethical approval) are uploaded.

The manuscript text is the "clean" version with all track changes accepted and no comment bubbles.

Conclusion

The manuscript is in its final stages. By implementing these focused revisions for clarity, precision, and language polish, you will significantly strengthen it for publication. The study provides valuable, actionable insights for improving PACU care in similar resource-limited settings.

**Do you want your identity to be public for this peer review?** For information about this choice, including consent withdrawal, please see our Privacy Policy

Reviewer #1: **Yes:** Dr. Temiloluwa Adefusi

Reviewer #2: **Yes:** Ali Afkhaminia

---

## [Author Response · Author response to Decision Letter 1]

15 Dec 2025

Response to Reviewers

Manuscript ID: PONE-D-25-59170

Title: Assessing the Accuracy and Educational Value of YouTube Videos on a Novel Regional Anesthesia Technique (PENG Block)

Journal: PLOS ONE

General Response to the Editor

We thank you and the reviewers for the time and effort devoted to the evaluation of our manuscript.

All comments raised by Reviewer #1 have been carefully addressed, and the manuscript has been revised accordingly. A detailed, point-by-point response is provided below, and all changes are highlighted in the revised manuscript with track changes.

During the revision process, we carefully reviewed the comments provided by Reviewer #2. We respectfully note that these comments appear to refer to a different clinical study involving patient-level data (e.g., PACU outcomes, BMI categories, Clavien–Dindo classification, critical events, and odds ratios), which are not part of our manuscript. Our study is a cross-sectional content analysis of YouTube videos related to the PENG block and does not include any clinical or patient data. We therefore believe that Reviewer #2’s comments may have been inadvertently assigned to our manuscript and kindly request your guidance on how to proceed.

All journal-specific requirements and policies have been carefully reviewed and addressed.

Kind regards,

Muhammed Emin Zora, MD

on behalf of all authors

Reviewer #1

Comment 1

Ensure to use the same wordings in the text as was used in the table to allow for uniformity. In the results section, regarding reference to the DISCERN scale, correct “55.6% were mild” to “55.6% were moderate.”

Response:

The terminology in the Results section has been revised to ensure consistency with the tables. The term “mild” has been corrected to “moderate” when referring to the DISCERN scale.

Comment 2

In the comparison between individuals and other sources, correct the paragraph to:

“Differences between videos from non-individual sources using metrics such as VPI, likes, comments and quality scores as compared to individual videos were not statistically significant.”

Response:

The Results section has been revised accordingly, and the suggested sentence has been incorporated verbatim.

Comment 3

In the discussion section; 1st paragraph, moderate quality does not align with poor quality.

Response:

The first paragraph of the Discussion section has been rephrased to align with the study findings by referring to the generally low-to-moderate or suboptimal quality of health-related YouTube content reported in previous studies.

Comment 4

In the last line of the 4th paragraph, it should be preferably put as:

“In future research, though using a single scale may provide an indication of overall quality, using multiple scales would enhance reliability by providing cross validation.”

Response:

The last sentence of the fourth paragraph in the Discussion section has been revised using the recommended wording.

Comment 5

The percentage figures quoted for JAMA, GQS and Modified DISCERN scales in the Results section did not tally with the table.

Response:

The Results section has been carefully revised to ensure full consistency between the text and Table 2. The percentage figures for JAMA, GQS, and Modified DISCERN now exactly match the tabulated data. No changes were made to the statistical analyses or tables.

Reviewer #2

Response:

The comments provided by Reviewer #2 appear to refer to a different clinical manuscript involving patient-level data and outcomes. As the present study is a YouTube-based content analysis without clinical or patient data, these comments do not apply to this manuscript. We kindly request editorial guidance regarding this issue.

---

## [Decision Letter · Decision Letter 1]

13 Jan 2026

Assessing the Accuracy and Educational Value of YouTube Videos on a Novel Regional Anesthesia Technique (PENG Block)

PONE-D-25-59170R1

Dear Dr. Zora,

We’re pleased to inform you that your manuscript has been judged scientifically suitable for publication and will be formally accepted for publication once it meets all outstanding technical requirements.

Kind regards,

Cheong Kim

Academic Editor

PLOS One

Additional Editor Comments (optional):

Reviewers' comments:

Reviewer's Responses to Questions

**Comments to the Author**

Reviewer #1: All comments have been addressed

2. Is the manuscript technically sound, and do the data support the conclusions?

Reviewer #1: Yes

3. Has the statistical analysis been performed appropriately and rigorously?

Reviewer #1: Yes

4. Have the authors made all data underlying the findings in their manuscript fully available?

Reviewer #1: Yes

5. Is the manuscript presented in an intelligible fashion and written in standard English?

Reviewer #1: Yes

Reviewer #1: I love that all the corrections noted have been effected. The article is easy to read and is interesting. Well done to the team.

**Do you want your identity to be public for this peer review?** For information about this choice, including consent withdrawal, please see our Privacy Policy

Reviewer #1: **Yes:** ADEFUSI TEMILOLUWA

---

## [Editor Report · Acceptance letter]

PONE-D-25-59170R1

PLOS One

Dear Dr. Zora,

I'm pleased to inform you that your manuscript has been deemed suitable for publication in PLOS One. Congratulations! Your manuscript is now being handed over to our production team.

Kind regards,

on behalf of

Dr. Cheong Kim

Academic Editor

PLOS One